# Size and Flexibility Define the Inhibition of the H3N2 Influenza Endonuclease Enzyme by Calix[n]arenes

**DOI:** 10.3390/antibiotics8020073

**Published:** 2019-06-03

**Authors:** Yannick Tauran, José Pedro Cerón-Carrasco, Moez Rhimi, Florent Perret, Beomjoon Kim, Dominique Collard, Anthony W. Coleman, Horacio Pérez-Sánchez

**Affiliations:** 1Laboratoire Multimatériaux et Interfaces CNRS UMR 5615, Université Lyon 1, 69622 Villeurbanne, France; anthony.coleman@univ-lyon1.fr; 2Laboratory for Integrated Micro-Mechatronic Systems (LIMMS)/CNRS-IIS UMI 2820, Institute of Industrial Science, The University of Tokyo, Tokyo 153-8505, Japan; bjoonkim@iis.u-tokyo.ac.jp; 3Bioinformatics and High Performance Computing (BIO-HPC) Research Group, Universidad Católica de Murcia (UCAM), 30107 Murcia, Spain; jpceron@ucam.edu; 4Institut National de la Recherche Agronomique (INRA), UMR 1319 Micalis, F-78350 Jouy-en-Josas, France; moez.rhimi@jouy.inra.fr; 5Institut de Chimie et Biochimie Moléculaires et Supramoléculaires, CNRS UMR 5246, Université Lyon 1, 69622 Villeurbanne, France; florent.perret@univ-lyon1.fr; 6Centre for Interdisciplinary Research on Micro-Nano Methods (CIRMM), Institute of Industrial Science, University of Tokyo, Tokyo 153-8505, Japan; 7Université Lille, CNRS, Centrale Lille, Institut Supérieur de l’Electronique et du Numérique (ISEN), University Valenciennes, UMR 8520-IEMN, F59000 Lille, France; collard@iis.u-tokyo.ac.jp

**Keywords:** enzyme inhibitors, calix[n]arene, endonuclease, anti-viral activity, H3N2 virus, molecular docking

## Abstract

Inhibition of H3N2 influenza PA endonuclease activity by a panel of anionic calix[n]arenes and β-cyclodextrin sulfate has been studied. The joint experimental and theoretical results reveal that the larger, more flexible and highly water-soluble sulfonato-calix[n]arenes have high inhibitory activity, with para-sulfonato-calix[8]arene, SC8, having an IC_50_ value of 6.4 μM. Molecular docking calculations show the SC8 can interact at both the polyanion binding site and also the catalytic site of H3N2 influenza PA endonuclease.

## 1. Introduction

Influenza A viruses are agents of infection (pathogens) that can be transmitted by skin contact or from aerosol droplets. The result is a seasonal respiratory disease, characterized by fever, respiratory problems, and headaches. Influenza may be fatal in cases of highly active strains or for weakened patients, including children, the elderly, and those suffering from pre-existing chronic diseases [1]. Although infection rates are generally around four million per year, influenza epidemics have reached a pandemic level in 1918, 1957, 1968, and 2009 [2]. The severity of epidemics in terms of virus type, subtype, and lineage have recently been reviewed [3]. The host reservoir for the Influenza A virus is found in waterfowl; however, transmission to humans is possible. Infection of swine is more common and it has been postulated that swine may be a reservoir for gene reassortment between different sub-forms of the virus [4]. As the viral genome is composed of eight negative-sense RNA segments (vRNA) switching of various segments can lead to new Influenza A strains, as exemplified by the 2009 H1N1 pandemic where segments of avian, swine, and human origin were present [5].

Current action against influenza infection involves vaccination; however, such treatment is ineffective in post-infection or against emergent viral types [6]. At the moment, the approved anti-viral agents target the M2 ion channel (responsible for vRNP) or the neuraminidase (NA) protein responsible for the release of new virons from infected cells. Resistance to the M2 targeting drugs is quite well known and NA-inhibitor resistant viruses are becoming known. More importantly, some H5N1 isolates encode resistance to both approaches [7]; indeed some strains show drug resistance without prior exposure to active pharmaceutical ingredients (APIs) [8]. Of the constituent parts of the viral ribonuclease protein, the polymerase PA unit containing the endonuclease fragment is of potential interest, as described by White [9] who described the structural and biochemical requirements for the development of influenza virus inhibitors. During the past years, a series of small-molecular inhibitors targeting this endonuclease have been developed based on small basic nitrogen heterocyclic structures [10,11].

A few months ago, the FDA approved Xofluza using Baloxavir Marboxil as a prodrug of baloxavir acid for inhibiting the viral PA polymerase unit [12]. In vitro assay has demonstrated a potent inhibition with an IC_50_ of 2.5 nM compared to the other inhibitors (Table 1). Clinical trials showed also a greater reduction in viral loads compared to the neuraminidase inhibitor molecule oseltamivir [13]. However, resistance mutation constituted a clear concern with resistant viruses detection before (baseline) and after treatment [14]. As a consequence, other classes of molecule that target the active site of the endonuclease should be further investigated.

With regard to the use of large three-dimensional molecules to target endonuclease, PA previous studies have included the use of fullerene derivatives to complex the active pocket [15]. Our previous work on Class II endonucleases showed the interest of large macrocycles to inhibit PdiL, Hind III, and two other endonucleases [16]. There exists a wide variation in the sequence of various endonuclease enzymes belonging to the three or four recognized classes of endonucleases and in particular the large class of type II enzymes. In fact, the core structural geometries and both the nucleic acid recognition site and the catalytic site have conserved sequences, the former is the RXXR sequence [17] (strongly analogous to the BBXB recognition site for glycosyl-aminoglycan binding [18]) and the latter is characterized by the PD/EXK sequence [19]. In Figure 1 below, we show the sequence of the H3N2 PA endonuclease used from White et al. [9] with potential nucleic acid binding sites highlighted in red, several sites containing multiple R/KXXR/K sequences can be observed. 

The PD pair are involved in binding the catalytic metal center, generally Mg^2+^, but also Mn^2+^ and other divalent cations except for Ca^2+^ [20]. In that framework, previous studies of anionic calix[n]arene binding to glycosaminoglycan binding sites in Heparin Co-Factor II [21], the prion protein [22,23], and other proteins associated with neurodegenerative diseases [24] have shown that binding strength increases with macrocycle size. For para-sulfonato-calix[n]arene to protein binding only in the case of the serum albumins is the smallest anion para-sulfonato-calix[4]arene favored [25]. From these results and previously observed Class II endonuclease inhibition [15], a panel of nine sulfonated calix[n]arenes and cyclodextrins were selected, see Scheme 1. Two outriders were included cyclodextrin sulfate and calix[4]arene di-phosphorus acid, the former is an anti-viral [26] and an inhibitor of certain endonucleases [15]. The latter presents phosphate groups, possible analogs to the phosphodiester groups of DNA. A major factor in the choice of potentially active molecules lies in the low toxicity of the anionic calix[n]arenes [27].

Then, blind docking method has been carried out to predict where the calix[n]arene interacted with endonuclease PA, as we expected interaction at the DNA holding site and not the active cleavage site. As stated by Mallipeddi [28], computational screening may offer a hit rate that is 10 times higher (1–5%) than random screening. Therefore, blind docking has been found out here particularly appropriated in our methodology to suggest the DNA binding groove site as one of the possible target sites.

## 2. Results

SC4a is presumed to be in the cone conformation [29] as is SC4c; however, SC4b may be present in a cone or 1,3 alternate, or even a partial cone structure. C4diP will be present in the cone conformation, as generally observed for this molecule [30]. With regard to the SC6 derivatives, we believe they will be present in the pleated “taco” form as previously observed for mono-functionalized para-H-calix-[6]-arenes, to obtain maximum length [31]. The question with regard to the SC8 systems is much more complex, structures varying from a flat roseate conformation, [32] through pleated loops [33] to a helical form with phenathroline resembling a DNA helix [34] are all known. It is interesting to note that the end to end distance at 2.0–2.2 nm is almost constant for all the conformations. The conformation appears to be entirely dependent on the nature of the species binding to SC8. Here, a twisted structure has been used in the calculations.

Recombinant gene expression was used to obtain the PA protein. The expression vector was constructed by S. Gaudon in the group of Cusack, EMBL Grenoble and kindly provided to us. The work produced a shorter but similar construct than that previously published by Dias [35]. The DNA coding for PA-Nter (residues 1–198) from A/Victoria/3/1975 (H3N2) (Accession number: CY121202.1) was cloned into a pET-M11 expression vector (EMBL). This vector was used to transfect *E. coli* BL21 and the protein was then expressed in LB medium over-night at 15 °C after induction with 0.1 mM isopropyl-β-thiogalactopyranoside (IPTG). Protein purification was undertaken using a metal affinity column (followed by gel filtration on a Superdex 200 column). The final concentration of protein was estimated at 0.5 mg/mL with a purity superior to 95% (Appendix A).

The digestion activity of the purified PA endonuclease was quantified using agarose gel electrophoresis. A concentration range of PA endonuclease (0.5 mg/mL to 0.5 µg/mL) was mixed with 0.5 µg of λ-DNA (Takara, Japan) in buffer at a final concentration of 20 mM Tris HCl pH 8, 100 mM NaCl, 1 mM MnCl_2_, 10 mM β-mercaptoethanol. After one hour of incubation at 37 °C, the samples were deposited on ethidium bromide treated agarose gel 0.6% and run during 90 minutes at 75 V. The gel was then scanned (Appendix A). 100 µM solutions of each macrocyclic molecules (Scheme 1) were mixed with 0.5 µg of buffered λ-DNA. This mixture was then combined with the PA endonuclease enzyme and incubated for 1 hour at 37 °C. The samples were deposited on agarose gel 0.6% previously mixed with ethidium bromide and run during 90 minutes at 75 V, after which the gel was scanned. Digestion activity was plotted as a function of inhibitor concentration by quantifying the intensity of the digested bands with ImageJ software. The IC_50_ values were calculated as in our work on endonuclease inhibition [8]. The intensity of λ-DNA bands was subtracted from negative control (λ-DNA lane) and then normalized from the positive control (λ-DNA and PA protein lane without inhibitor).

The results are given in Figure 2 for the inhibition at 100 µM for each macrocycle and in Figure 3 for the four active inhibitor molecules for varying concentrations. IC_50_ values are summarized in Table 1, full data is provided in the Appendix A.

The three dimensional structure of the PA endonuclease domain of the H3N2 Influenza strain has been determined by White et al. [9], shown below in Figure 4 with a charge coded surface and provided a base to carry out blind docking studies and more refined quantum mechanical calculations on the interaction between the most effective inhibitor SC8a and the potential binding hot spots located across the enzyme surface. The presence of a sulfate anion in the recognition site was of use in fixing the initial position of the anion.

Blind docking calculations for the top poses of SC8a are summarized in Figure 5, while Figure 6 illustrates the main interacting residues for top SC8a pose resulting from blind docking calculations into the PA cavity. Details on the main interacting residues for this pose are shown in Table 2.

Quantum mechanical calculations performed within the density functional theory framework confirm the pivotal role played by the region of the protein in close contact to the top pose of SC8a, whose scoring function value is differentiated from the rest of poses (see Appendix A). More specifically, Lys34 and Lys115 residues largely contribute to the stabilization of the binding site with associated pairwise energy of −274.6 kcal/mol and −322.6 kcal/mol, respectively, through electrostatic interactions, shape complementarity and a network of hydrogen bonds. His41 residue presents also a minor contribution with –18.1 kcal/mol compared to the global binding energy. It is also noticeable that the large positive energy with Glu114, which is predicted at 248.6 kcal/mol. It is worth noting that the electronic attraction/repulsion forces play a pivotal role in pairwise energies so that the predicted repulsion with that specific residue is the logical consequence of the close location of negatively charged species. However, the final balance significantly stabilizes the binding site by the other attractive contacts, especially by interacting with Lys34 and Lys115, as discussed above. For the records, the other four poses show other probable binding sites for SC8a However, all distant from the catalytic site and so may be presumed to have no effect on inhibition of the endonuclease.

## 3. Discussion

The amino acid sequences for the construct used in the inhibition studies and the docking studies are not identical, this does not affect comparison as the 130–140 sequence that is essential for binding is fully conserved in the two structures. The two sequences are given below in Figure 6.

Comparison of the two sequences shows a very high degree of conservation between the construct and the crystallographic protein and in particular that there is full conservation of the binding site found in the docking experiment.

The docking experiments show that there are five possible binding sites with the Pose 1 preferred binding site interacting with the Lys 34, Tyr 130 Lys 134, and Lys 137 residues. The local binding environment is shown below in Figure 7A. The four other possible binding sites which would have no effect on the endonuclease inhibition have been removed. Figure 7B shows the interacting residues at the pharmacophore pocket as defined by Credille et al. [39].

The inhibition experiments hint that both the size of the macrocyclic ring and the substitution have effects on the binding, as probably does conformational flexibility. Hence, neither SC4 and its derivatives nor C4diP show any inhibitory effects, this would imply that binding to more than one amino acid residue is required for enzyme inhibition. The solid-state structures of SC4 with Lys [40] or Arg [41] confirm that this system only binds a single amino acid. For the larger systems, SC6 and SC8 inhibitory effects occur for the SCNa molecules; this would be in line with binding to pharmacophore pockets 1 and 6, as shown in Figure 7 below. The lack of flexibility of the CD sulfate polyanion will rule out interaction across the two pockets, and thus it is inactive. The larger SC8a anion can bind to the K34, Y130, K134, and K137 residues while the smaller SC6a will probably bind only to two of these residues. This reduction in binding will be responsible for the increased IC_50_ 6.4µM observed for SC8a compared to that of 11.2µM for SC6a.

Of some interest is the effect of the O-alkyl sulfonate substituent group on the inhibitory effect. It would be expected if the binding was uniquely electrostatic in nature that the addition of an additional negative charge, should increase the inhibitory effect.

The IC_50_ values decrease from 11.2 µM for SC6a to greater than 100 µM for SC6b and SC6c, and 6.4 µM, 14 µM, and 37 µM for SC8a, SC8b and SC8c respectively, from Figure 4 it can be seen that conformation of the calix[n]arene ring presents the hydroxyl in close spatial proximity thus substitution at the phenolic face will cause steric crowding and repulsion for the SCNc series of molecules. For the SCNb series, the difference can be explained by the difference in the alkyl-sulfonate group binding to the positive amino acids as compared to the aromatic sulfonate group binding.

In terms of inhibitor design, the anionic *para*-sulfonato-calix[n]arenes represent a different class of molecules than more classical pharmaceutical inhibitors, [42,43] in that they are strongly anionic large three-dimensional macrocycles as compared to relatively small basic nitrogen heterocyclic structures of the classic inhibitors. The binding experiments are based on analysis of the peptide sequence of the protein so as to verify the presence of suitable binding sites for this type of supramolecular system. Coleman’s group has already demonstrated that this type of sequence analysis based on *para*-sulfonato-calix[n]arene derivatives as a protein ligand enables the identification of other types of pharmaceutical targets, such as the Tau protein, which is involved in neurodegenerative diseases [24].

## 4. Materials and Methods

### 4.1. Synthesis and Characterization of Calix[n]arenes

All chemicals were purchased from ACROS Organics or Sigma Aldrich and used without further purification. Solvents were of chemical grade and were used without any purification.

The starting materials, calix[n]arenes were prepared by deterbutylation of para-tert-butylcalix[n]arenes, using the procedure described by Gutsche [44].

Para-sulfonato-calix[n]arenes SC4a, SC6a, and SC8a have been synthesized according to the procedure described by Coleman et al. [45].

Calix[n]arenes propane-3-sulfonic acid SC4b, SC6b and SC8b were synthesized by the reaction of these calix[n]arenes with propane-1,3-sultone described for para-tert-butyl-calix[6]arene by Shinkai et al. [46].

Para-sulfonato-alkylcalix[n]arenes propane-3-sulfonic acid SC4c, SC6c, and SC8c have been prepared following the procedure described by Hwang et al. [47] and adapting the procedure of Shinkai et al. [48]

Calix[4]arene dihydroxyphosphonic acid (C4diP) was synthesized by the method of Markovsky and Kalchenko [49].

All the physical characteristics of the synthesized calixarenes correspond to the literature values.

Sulfated β-cyclodextrin (CD) has been purchased from Sigma-Aldrich (Saint Louis, MO, USA).

### 4.2. Over-Expression and Purification of PA Endonuclease

The vector of PA endonuclease has been kindly sent by Dr. Kuzack from EMBL upon our request.

The DNA coding for PA-Nter (residues 1–198) from A/Victoria/3/1975 (H3N2) (Accession number: CY121202.1) was cloned into a pET-M11 expression vector (EMBL).

This vector was used to transform *E. coli* BL21 (DE3) RIL CodonPlus strain (Stratagene). The protein was expressed in LB medium over-night at 15 °C after induction with 0.1 mM isopropyl-β-thiogalactopyranoside (IPTG). The protein was first purified by an immobilized metal affinity column (IMAC) and then by gel filtration on a Superdex 200 column (GE Healthcare, Japan).

The final concentration of protein was estimated at 0.5 mg/mL with a purity superior to 95% (Appendix A). The purity of the protein was assessed by SDS PAGE. One µg of purified PA endonuclease was loaded (2 µL at 500 µg/mL) on the polyacrylamide gel and has revealed a unique and intense band by Coomassie staining. Since the limit of detection of Coomassie stain is 0.05 µg [50], we assumed that the purity of the endonuclease is superior to 95%.

### 4.3. PA Endonuclease Activity

The digestion activity of the purified PA endonuclease has been quantified on agarose gel electrophoresis. A concentration range of PA endonuclease (0.5 mg/mL until 0.5 µg/mL) has been mixed to 0.5 µg of λ-DNA (Takara, Japan) in a buffer at a final concentration of 20 mM Tris HCl pH 8, 100 mM NaCl, 1 mM MnCl2, 10 mM β-mercaptoethanol. After one hour of incubation at 37 °C, the samples have been deposited on agarose gel 0.6% previously mixed with Ethidium bromide and run over 90 min at 75 V. The gel was then scanned on ChemiDoc XRS system (Bio-Rad) (Appendix A)

### 4.4. PA Endonuclease Inhibition Assay for Macrocyclic Molecules

100 µM of each macrocyclic molecules β-CDsul, SC[n]a, SC[n]b, and SC[n]c (Figure 1) have been mixed to 0.5 µg of λ-DNA in a buffer at final concentration of 20 mM Tris HCl pH 8, 100 mM NaCl, 1 mM MnCl2, 10 mM β-mercaptoethanol.

The mixture was then mixed with the PA endonuclease enzyme at 0.5 mg/mL and incubated for 1 h at 37 °C.

The samples have been deposited on agarose gel 0.6% previously mixed with ethidium bromide and run over 90 min at 75 V. The gel was then scanned on the ChemiDoc XRS system (Bio-Rad).

The digestion activity is then plotted as a matter of inhibitor concentration by quantifying the intensity of the digested bands with ImageJ software.

### 4.5. Molecular Modeling Calculations

We performed molecular modeling studies in order to obtain information at the atomic level about the interactions between the different inhibitors studied and PA endonuclease. For such purpose, we employed docking calculations, which consist in the computer simulation of the energetics of the protein-ligand interactions and thus the estimation of binding affinity values and the structure the protein-ligand complexes adopt when the protein binds with the studied inhibitors.

Predictions obtained by docking simulations can reveal useful information [51] about the presence or absence of protein-ligand interactions and about how these interactions are established (electrostatic, van der Waals, hydrogen bonds, hydrophobic, etc.) and concerning which residues of the protein are involved. A representative protein X-ray crystal structure for each protein formulation was chosen at the Protein Data Bank (PDB) database (http://www.rcsb.org/pdb). For PA endonuclease, the structure with PDB identifier 4E5G was selected.

The full-atom models of the protein used in this study was prepared from the raw PDB structure 4E5G by removing water molecules, adding hydrogen atoms, assigning the ionization states of the amino acids by using Protein Preparation Wizard module as implemented in Maestro (Schrödinger Release 2016-4: Maestro, Schrödinger, LLC, New York, USA) [52]. Protonation states of all side chains were subsequently defined using PROPKA3.1. Partial charges over all atoms were finally assigned within the AMBER99 force field scheme as implemented in AmberTools (AMBER 2017, University of California, San Francisco) [53]. The docking of the inhibitors to the prepared protein structure model of PA endonuclease and the detailed binding energy calculations were performed with the Lead Finder v 1.1.10 software [54] using default configuration parameters. The size of the grid box for ligand docking was set to extend 60 A° in each direction from the geometric center of each individual docking simulation. The dG-score produced by Lead Finder was taken as the predicted value of the ligand binding energy. Only the top-ranked poses were used for structural and energy analyses.

We performed a blind docking approach [55] where multiple docking runs started around geometric centers of all residues within the selected threshold. A histogram with the resulting distribution of binding energies is shown in Appendix A, and the pose with the lowest energetic value is selected as the one that represents the experimental results.

For the top pose obtained in blind docking, pairwise energies are computed at the M06-2X/6-311+G(d,p) level of theory [56,57] with the basis set superposition error (BSSE) as established in the well-known counterpoise method [58]. All quantum mechanical calculations were performed by using Gaussian09 [59]. More details on the molecular modeling calculation can be found in a previous paper [60].

## 5. Conclusions

In conclusion, we demonstrated that large flexible *para*-sulfonato-calix[n]arenes could inhibit in vitro H3N2 PA endonuclease. Combining the methodology of experimental inhibition studies and theoretical docking work has given us insights on the mechanism of endonuclease inhibition by para-sulfonato-calix[8]arene. The larger SC8a anion can bind to the K34, Y130, K134, and K137 residues of the DNA binding pharmacophore pocket and also interact at the catalytic site. The flexibility of calix[8]arene might also present an advantage against resistant virus strains, which has been mentioned as one of the only concern in the recent PA endonuclease inhibitor molecule Xofluza approved by FDA. Finally, the low animal toxicity of this class of molecules opens up the possibility of the design of totally novel active pharmaceutical ingredients (APIs) based on atypical anionic molecules.

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
