# Peer review of "Size and Flexibility Define the Inhibition of the H3N2 Influenza Endonuclease Enzyme by Calix[n]arenes"

_antibiotics, 2019, doi:10.3390/antibiotics8020073_

Round 1
Reviewer 1 Report
This manuscript describes the experimental observation of inhibition of influenza endonuclease by several calixarene derivatives, along with computational calculations of docking of the most potent derivative with the enzyme.
The experimental work seems carefully performed, and reveals that the interaction is quite specific: only some of the calixarenes show significant inhibition of the enzyme. The calixarenes are all known, heavily sulfonated derivatives: this reduces the impact of the work because there is no new chemistry, and perhaps a less than ideal panel of controls, e.g. it would be nice to see the data even if only for the SC8a analogue lacking sulfonate groups. Nonetheless, the results do seem to establish that the calix[8]arenes are better inhibitors than the smaller ring [4]- or [6]-analogues.
Likewise, the docking studies for SC8a begin to reveal the nature of the interaction with the protein; however this section would be more convincing if it included more analogues, especially a negative control compound, and reproduced the order of potency of the observed effective inhibitors. My feeling is that some of the deductions from the modelling study may have been over-interpreted in light of the lack of comparator results.
Overall, this seems like a preliminary study but does reveal some positive inhibition data that may be applicable to tackling the influenza virus; the target alone will make this paper interesting to a wide range of scientists. I would encourage the authors to consider toning down some of the deductions where they are not well supported by appropriate control experiments, but with that proviso I think this work meets the standards required for publication in J. Antibiot..
Some specific points:
Line 48: there is no real review of any “traditional” drugs that target the PA enzyme; this would be useful to the non-expert, and provide comparators for the efficacy of e.g. SC8a. Lines 255-7 suggest some are known.
Line 111: bracket opens but no close
Line 112: “purity /superior to /of 95%” what does this mean?
Line 150: Figure 4: what is docked in the enzyme (the sticks)? Define.
Line 230-231: “neither… or…show no…” should be neither….nor….show any….
Lines 236-239: this illustrates an over-stretch in interpreting the data. The authors suggest a reason why binding of SC6a is different from that of SC8a, yet this can only really be substantiated by modelling SC6a. In this case, I would expect dropping from 4 binding interactions to 2 would create a much larger effect than the 2-fold difference in IC50 that is observed experimentally.
Author Response
General response:
The reviewer stated that “this work meets the standards required for publication in J. Antibiot.”
However, the reviewer raised an important point about the conclusion that may have been over-interpreted in light of the lack of comparator results.
About the comparator point, we have addressed it in details in the point 1).
Concerning the conclusion, we have changed it accordingly to fit better with the obtained results: (Line 372)
In conclusion, we demonstrated that large flexible para-sulphonato-calix[n]arenes can inhibit in vitro H3N2 PA endonuclease. Combining the methodology of experimental inhibition studies and theoretical docking work has given us insights on the mechanism of endonuclease inhibition by para-sulphonato-calix[8]arene. The larger SC8a anion can bind to the K34, Y130, K134 and K137 residues of the DNA binding pharmacophore pocket and also interact at the catalytic site. The flexibility of calix[8]arene might also presents an advantage against resistant virus strains which has been mentioned as one of the only concern in the recent PA endonuclease inhibitor molecule Xofluza approved by FDA. Finally, the low animal toxicity of this class of molecules opens up the possibility of the design of totally novel Active Pharmaceutical Ingredients (APIs) based on atypical anionic molecules.
Some specific points:
1) Line 48: there is no real review of any “traditional” drugs that target the PA enzyme; this would be useful to the non-expert, and provide comparators for the efficacy of e.g. SC8a. Lines 255-7 suggest some are known.
Response:
A paragraph has been added about the only available drug that inhibits PA Endonuclease whose use has been authorized by FDA a few months ago (Line 47):
Of the constituent parts of the viral ribonuclease protein, the polymerase PA unit containing the endonuclease fragment is of potential interest, as described by White [9] who described the structural and biochemical requirements for the development of Influenza Virus Inhibitors. During the past years, a series of small-molecular inhibitors targeting this endonuclease have been developed based on small basic nitrogen heterocyclic structures [10, 11].
A few months ago, the FDA approved Xofluza using Baloxavir Marboxil as a prodrug of baloxavir acid for inhibiting the viral PA polymerase unit [12]. In vitro assay has demonstrated a potent inhibition with an IC50 of 2.5 nM compared to the other inhibitors (Table 1). Clinical trials showed also a greater reduction in viral loads compared to the M2 ion channel inhibitor molecule Oseltamivir [13]. However, resistance mutation constituted a clear concern with resistant viruses detection before (baseline) and after treatment [14]. As a consequence, other classes of molecule that target the active site of the endonuclease should be further investigated.
For better comparison, Table 1 has been significantly implemented with the addition of molecules that inhibit different Influenza virus pharmaceutical targets (M2 channel, Neuramidase and PA Endonuclease) (Line 156):
Table 1. IC50 values for inhibition of Influenza virus pharmaceutical targets. Commercially available drugs that inhibit M2 channel, Neuramidase and PA Endonuclease targets are in green. PA endonuclease inhibitors assessed only by in vitro assays are in yellow. Molecules investigated in this study are color coded in red to show inhibition below 20 µM, and in grey for none inhibition observed under 100 µM.
2) Line 111: bracket opens but no close
Response: The typo has been corrected.
3) Line 112: “purity /superior to /of 95%” what does this mean?
Response: The sentence has been corrected (Line 129).
The final concentration of protein was estimated at 0.5 mg/mL with a purity superior to 95% (Figure S1).
An explanation about the estimation of protein purity is given in the material of method (Line 317). The purity of protein was assessed by SDS PAGE. 1 µg of purified PA endonuclease was loaded (2 µL at 500 µg/mL) on the poly-acrylamide gel and has revealed a unique and intense band by Coomassie staining. Since the limit of detection of Coomassie stain is 0.05µg [41], we assumed that the purity of the endonuclease is superior to 95 %.
4) Line 150: Figure 4: what is docked in the enzyme (the sticks)? Define.
Response: Sticks on figure 4 correspond to SC8a.
The caption of figure 4 has been corrected (Line 169):
Figure 4. The three dimensional structure of the PA Endonuclease domain of the H3N2 Influenza strain with the top pose obtained from Blind Docking calculations for SC8a (in sticks), the surface is color coded with positive charge shown in blue and negative charge shown in red. The SC8a structure used is CCDC 970626 from [34]
5) Line 230-231: “neither… or…show no…” should be neither….nor….show any….
Response: The sentence has been corrected (Line 255):
Hence, neither SC4 and its derivatives nor C4diP show any inhibitory effects, this would imply that binding to more than one amino acid residue is required for enzyme inhibition.
6) Lines 236-239: this illustrates an over-stretch in interpreting the data. The authors suggest a reason why binding of SC6a is different from that of SC8a, yet this can only really be substantiated by modelling SC6a. In this case, I would expect dropping from 4 binding interactions to 2 would create a much larger effect than the 2-fold difference in IC50 that is observed experimentally.
Response: In fact, there will be the same binding for the 6 and 8 at DNA binding groove as the expected structure of the calix-6 is such that four sulphonic groups can bind. The difference which is admittedly quite small will come from the coverage at the catalytic site where sulphonic group to Mn binding is not necessarily strong.

Reviewer 2 Report
Tauran et al., presented calix[n]arenes as H3N2 influenza PA endonuclease inhibitors via experimental and theoretical studies. There is minimal novelty in the compounds tested as well as the way study has been performed. Docking has been performed with minimal knowledge of the method. The study is flawed and results are not reliable. For the above mentioned reasons as well as for the ones given below, I do not recommend the article for publication in current form.
For Table 1, can the authors also presence a known inhibitor activity?
Why did the authors perform blind docking studies? There is enough information on endonuclease binding site. e.g. Refer to the 'endonuclease' section in 'influenza drug discovery CTMC 2014 1875-1889'.
How did you generate initial conformations for calix[n]arenes? Ligand conformations is a crucial part in docking studies. Refer CTMC 2014 1875-1889.
Line 165: It is concerning that the interaction energy is positive for Glu114.
Lines 259-260 - What does this sentence mean? Isn't the target influenza? What does Neurodegenerative diseases mean?
Line 49: Cite references for endonuclease site
Author Response
General response:
Reviewer 2 found two fundamental problems (Novelty of the study and Methodology) with the manuscript:
- Concerning the novelty of the study, we revealed in this preliminary study that large suphonate calix[n]arenes inhibit in vitro endonuclease PA. This new class of molecule presents an interest in the development of anti-influenza agents, especially regarding the resistance virus strains concern. To emphasize better our message, we improved the introduction by adding a paragraph about a new PA endonuclease inhibitor that has just been approved by FDA (Line 49). A table has been added comparing the inhibition activity of the sulphonated calix[n]arene against other molecules (commercially available or not) that have been found to inhibit Influenza virus (Table 1). Finally, the conclusion has been modified to fit better with the results obtained (Line 372).
- About the use of molecular docking in the methodology, the answer has been addressed in details in the following points 2, 3 and 4.
After having proceeded to a number of changes in the article, now we feel that this work meets the standards required for publication in Antibiotics Journal.
1) For Table 1, can the authors also presence a known inhibitor activity?
Response:
Table 1 has been significantly implemented with the addition of commercially available molecules that inhibit different Influenza virus pharmaceutical targets (M2 channel, Neuramidase and PA Endonuclease):
Table 1. IC50 values for inhibition of Influenza virus pharmaceutical targets. Commercially available drugs that inhibit M2 channel, Neuramidase and PA Endonuclease targets are in green. PA endonuclease inhibitors assessed only by in vitro assays are in yellow. Molecules investigated in this study are color coded in red to show inhibition below 20 µM, and in grey for none inhibition observed under 100 µM.
2) Why did the authors perform blind docking studies? There is enough information on endonuclease binding site. e.g. Refer to the 'endonuclease' section in 'influenza drug discovery CTMC 2014 1875-1889'.
Response:
Blind docking calculations were used in order to show to which other additional sites, apart from the known endonuclease binding site, the different studied ligands might bind to. This information is relevant since the number of potential interacting sites can inform about the capacity of endonuclease to carry more than one calixarene molecule, and the higher this number the more inhibitory capacity for the ligand.
As stated by Mallipeddi in the 2014 CTMC article [28] computational screening may offer a hit rate that is
10 times higher (1%–5%) than random screening, thus it is logical here where a new binding site - the DNA binding groove - to confirm by Blind Docking that the site is correctly targeted and hence that the current work is in agreement with the body of previous studied on binding to the catalytic site.
In this case, the blind docking leads also to the possibility that the calix[8]arene skeleton is large enough to also affect the active site.
A paragraph has been added in the introduction to explain better the use of blind docking in our methodology:
(Line 103)
Then, blind docking method has been carried out to predict where the calix[n]arene interacted with Endonulease PA, as we expected interaction at the DNA holding site and not the active cleavage site. As stated by Mallipeddi [28] computational screening may offer a hit rate that is 10 times higher (1%–5%) than random screening. Therefore, blind docking has been found out here particularly appropriated in our methodology to confirm that the DNA binding groove site was correctly targeted.
Reference:
[28] Mallipeddi, P.L.; Kumar, G.; White, S.W.; Webb, T.R. Recent advances in computer-aided drug design as applied to anti-influenza drug discovery. Curr Top Med Chem. 2014, 14, 1875-1889.
3) How did you generate initial conformations for calix[n]arenes? Ligand conformations is a crucial part in docking studies. Refer CTMC 2014 1875-1889.
Response:
The initial conformation was taken from a solid-state structure involving binding to a cationic organic diamine as was in the suspected DNA binding groove [34].
As a starting point for docking calculations, the structures of all calix[n]arenes were taken from Cambridge Crystallographic database CCDC 970626.
The reference is given in the caption of figure 4 :
Figure 4. The three dimensional structure of the PA Endonuclease domain of the H3N2 Influenza strain with the top pose obtained from Blind Docking calculations for SC8a (in sticks), the surface is color coded with positive charge shown in blue and negative charge shown in red. The SC8a structure used is CCDC 970626 from [34]
Reference :
[34] Leśniewska, B.; Coleman, A.W.; Tauran, Y.; Perret F.; Suwińska K. Pseudopolymorphs - a variety of self-organization of para-sulphonato-calix[8]arene and phenanthroline in the solid state. CrystEngComm. 2016, 18, 8858-8870
4) Line 165: It is concerning that the interaction energy is positive for Glu114.
Response:
At the pH in proximity to a polysulphonic acid the Glu is likely to be protonated and H-bond – indeed in the review by Perret for RSC (reference below) the binding constant is high, hence this binding is not unexpected.
Reference :
Perret, F.; Coleman, A.W. Biochemistry of anionic calixarenes. Chem. Commun. 2011, 47(26), 7303-7319
A sentence has been added in the text to clarify this point (Line 186):
It is worth noting that the electronic attraction/repulsion forces play a pivotal role in pairwise energies, so that the predicted repulsion with that specific residue is the logical consequence of the close location of negatively charged species. However, the final balance significantly stabilizes the binding site by the other attractive contacts, especially by interacting with Lys34 and Lys115, as discussed above.
5) Lines 259-260 - What does this sentence mean? Isn't the target influenza? What does Neurodegenerative diseases mean?
Response: For sake of clarity, the sentence has been rephrased (Line 283):
Coleman’s group has already demonstrated that this type of sequence analysis based on para-sulphonato-calix[n]arene derivatives as a protein ligand was efficient to recognize other types of pharmaceutical targets such as Tau protein involved in neurodegenerative diseases [24].
6) Line 49: Cite references for endonuclease site
Response:
Sentences including the reference for the endonuclease site has been added in the text (Line 47) :
Of the constituent parts of the viral ribonuclease protein, the polymerase PA unit containing the endonuclease fragment is of potential interest, as described by White [9] who described the structural and biochemical requirements for the development of Influenza Virus Inhibitors. During the past years, a series of small-molecular inhibitors targeting this endonuclease have been developed based on small basic nitrogen heterocyclic structures [10, 11].
And line 68:
In Figure 1 below, we show the sequence of the H3N2 PA endonuclease used from White et al [9] with potential nucleic acid binding sites highlighted in red, several sites containing multiple R/KXXR/K sequences can be observed.

Reviewer 3 Report
The present manuscript reports the ability of a variety of anionic calix[n]arenes and β-cyclodextrin sulphate to inhibit the activity of H3N2 Influenza virus PA endonuclease. The most active molecule showed a IC50 value in the single digit micromolar order of concentration. The results of the inhibitory activities were the interpreted on the basis of the structural features of the molecules and on molecular docking simulations. Overall, the work is interesting, focusing on inhibitors of an important antiviral therapeutic target. The manuscript is well organized with an appropriate discussion of the obtained findings. I recommend its publication after some revisions, as described below.
- the document needs some language polishing; commas are missing or wrongly included in some sentences, which make difficult the reading of some parts of the manuscript.
- line 22, italicize “para” in “para-sulphonato-calix[8]arene” and throughout the manuscript.
-line 42, “ineffective post-infection” should be corrected to “ineffective in post-infection”
-line 45, “NA resistance viruses” should be corrected to “NA-inhibitor resistant viruses” or “viruses resistant to NA inhibitors”.
-line 88, “phosphor-ester groups” should be corrected to “phosphodiester groups”
A standard PA endonuclease inhibitor should be subjected to inhibition assays to have a reference for a better discussion about the potency of the compounds.
Author Response
General response:
Reviewer 3 has stated that he “recommend its publication after the following revisions”.
1) - the document needs some language polishing; commas are missing or wrongly included in some sentences, which make difficult the reading of some parts of the manuscript.
Response: Language polishing has been performed by one of us, Dr. Anthony W. Coleman, a native English graduate of the University of Oxford. Missing or wrong commas have been corrected (Lines 35, 39, 45, 48, 57, 89, 111, 162, 185, 292, 341 and 345).
2) - line 22, italicize “para” in “para-sulphonato-calix[8]arene” and throughout the manuscript.
Response: The sentence has been corrected (Line 22)
The joint experimental and theoretical results reveal that the larger, more flexible and highly water soluble sulphonato-calix[n]arenes have high inhibitory activity, with para-sulphonato-calix[8]arene, SC8, having an IC50 value of 6.4 μM.
3) -line 42, “ineffective post-infection” should be corrected to “ineffective in post-infection”
Response: The sentence has been corrected (Line 42)
Current action against Influenza infection involves vaccination, however such treatment is ineffective in post-infection or against emergent viral types [6].
4) -line 45, “NA resistance viruses” should be corrected to “NA-inhibitor resistant viruses” or “viruses resistant to NA inhibitors”.
Response: The sentence has been corrected (Line 45)
Resistance to the M2 targeting drugs is quite well known, and NA-inhibitor resistant viruses are becoming known.
5) -line 88, “phosphor-ester groups” should be corrected to “phosphodiester groups”
Response: The sentence has been corrected (Line 95)
The latter presents phosphate groups, possible analogs to the phosphodiester groups of DNA.
6) A standard PA endonuclease inhibitor should be subjected to inhibition assays to have a reference for a better discussion about the potency of the compounds.
Response:
A paragraph has been added about the only available drug that inhibits PA Endonuclease whose has been authorized by FDA a few months ago (Line 49):
Of the constituent parts of the viral ribonuclease protein, the polymerase PA unit containing the endonuclease fragment is of potential interest, as described by White [9] whom described the structural and biochemical requirements for the development of Influenza Virus Inhibitors. During the past years, a series of small-molecular inhibitors targeting this endonuclease have been developed based on small basic nitrogen heterocyclic structures [10, 11].
A few months ago, the FDA approved Xofluza using Baloxavir Marboxil as a prodrug of baloxavir acid for inhibiting the viral PA polymerase unit [12]. In vitro assay has demonstrated a potent inhibition with an IC50 of 2.5 nM compared to the other inhibitors (Table 1). Clinical trials showed also greater reduction in viral loads compared to the M2 ion channel inhibitor molecule Oseltamivir [13]. However, resistance mutation constituted a clear concern with resistant viruses detection before (baseline) and after treatment [14]. As a consequence, other classes of molecule that target the active site of the endonuclease should be further investigated.
For better comparison, Table 1 has been significantly implemented with the addition of commercial and not commercial molecules that inhibit different Influenza virus pharmaceutical targets (M2 channel, Neuramidase and PA ndonuclease)(Line 156):
Table 1. IC50 values for inhibition of Influenza virus pharmaceutical targets. Commercially available drugs that inhibit M2 channel, Neuramidase and PA Endonuclease targets are in green. PA endonuclease inhibitors assessed only by in vitro assays are in yellow. Molecules investigated in this study are color coded in red to show inhibition below 20 µM, and in grey for none inhibition observed under 100 µM.

Round 2
Reviewer 2 Report
Calix[n]arenes as Endonuclease inhibitors by Tauran et.al., meets the standards of 'Antibiotics' journal after the following revisions
Major issue:
Line 165: It is still a major concern that the interaction energy is positive for Glu114. It raises questions about the way blind docking was done. Algorithm might have given a random docked pose, which needs to be refined. Did you energy minimize the docked pose? Alternate way to overcome this is to do focussed docking i.e. use the predicted binding site residues from blind docking and redo docking only at this site.
Minot issues:
Lines 106-107: Blind docking confirms DNA binding groove site is an overstatement.
Lines 246-247: Sentences needs to be rephrased - ....four binding to quite removed sites....
Lines 281-284: Sentences needs to be rephrased - ....protein ligand was efficient to recognize....
Author Response
Our point-by-point responses to reviewer 2 are detailed below. All the revisions in the article have been highlighted in yellow.
After having addressed these minor revisions in the article, now we feel that this work meets the standards required for publication in Antibiotics Journal.
Open Review
(x) I would not like to sign my review report
( ) I would like to sign my review report
English language and style
( ) Extensive editing of English language and style required
( ) Moderate English changes required
(x) English language and style are fine/minor spell check required
( ) I don't feel qualified to judge about the English language and style
Yes | Can be improved | Must be improved | Not applicable | |
Does the introduction provide sufficient background and include all relevant references? | (x) | ( ) | ( ) | ( ) |
Is the research design appropriate? | ( ) | ( ) | (x) | ( ) |
Are the methods adequately described? | ( ) | (x) | ( ) | ( ) |
Are the results clearly presented? | ( ) | (x) | ( ) | ( ) |
Are the conclusions supported by the results? | ( ) | (x) | ( ) | ( ) |
Comments and Suggestions for Authors
Calix[n]arenes as Endonuclease inhibitors by Tauran et.al., meets the standards of 'Antibiotics' journal after the following revisions
Major issue:
Line 165: It is still a major concern that the interaction energy is positive for Glu114. It raises questions about the way blind docking was done. Algorithm might have given a random docked pose, which needs to be refined. Did you energy minimize the docked pose? Alternate way to overcome this is to do focussed docking i.e. use the predicted binding site residues from blind docking and redo docking only at this site.
Answer:
We describe in the manuscript that “… It is also noticeable the large positive energy with Glu114..., and it is the logical consequence of the repulsive interactions between both negatively charged species.”, which refers mostly to repulsions between oxygen atoms. However, in our blind docking procedure, we performed many different focused docking simulations around all resides to exhaustively sample whole protein surface and detect most probable interactions sites. We should highlight that on each individual docking simulation from the blind docking approach, we, of course, performed energy minimization. Thus, at the end the obtained poses are ranked according to their total interaction score, which is the total sum of favorable plus disfavorable interactions. In the case of the mentioned pose, the large positive energy of Glu114 is fully balanced by favorable electrostatic and hydrogen bond interactions with the other mentioned residues. Since the objective of the paper is not to describe in detail the blind docking approach, for such purpose the reviewer can read our most recent paper that uses it in Nature Chemical Biology.
For the reader, the reference has been also supplemented in our Material and Method section (Line 360):
More details on the Molecular Modeling Calculation can be found in a previous paper of one of us [60].
Reference:
[60] Tapia-Abellán, A.; Angosto-Bazarra, D.; Martínez-Banaclocha, H.; de Torre-Minguela, C.; Cerón-Carrasco, J.P.; Pérez-Sánchez, H.; Arostegui, J.I. & Pelegrin, P. MCC950 closes the active conformation of NLRP3 to an inactive state. Nat. Chem. Biol. 2019, 15, 560–564.
Minor issues:
Lines 106-107: Blind docking confirms DNA binding groove site is an overstatement.
Instead of “Therefore, blind docking has been found out here particularly appropriated in our methodology to confirm that the DNA binding groove site was correctly targeted.”
Answer: Therefore, blind docking has been found out here particularly appropriated in our methodology to suggest the DNA binding groove site as one of the possible target sites.
Lines 246-247: Sentences needs to be rephrased - ....four binding to quite removed sites....
Instead of “The docking experiments show that there are five possible binding sites with the Pose 1 preferred binding site interacting with the Lys 34, Tyr 130 Lys 134 and Lys 137 residues, the other four binding to quite removed sites which would have no effect on the endonuclease inhibition, the local binding environment is shown below in Figure 7a”
Answer: The docking experiments show that there are five possible binding sites with the Pose 1 preferred binding site interacting with the Lys 34, Tyr 130 Lys 134 and Lys 137 residues. The local binding environment is shown below in Figure 7a. The four other possible binding sites which would have no effect on the endonuclease inhibition, have been removed.
Lines 281-284: Sentences needs to be rephrased - ....protein ligand was efficient to recognize....
Instead of “Coleman’s group has already demonstrated that this type of sequence analysis based on para-sulphonato-calix[n]arene derivatives as a protein ligand was efficient to recognize other types of pharmaceutical targets such as Tau protein involved in neurodegenerative diseases [24].”
Answer: Coleman’s group has already demonstrated that this type of sequence analysis based on para-sulphonato-calix[n]arene derivatives as a protein ligand enables the identification of other types of pharmaceutical targets such as Tau protein involved in neurodegenerative diseases [24].
